# Safety, Efficacy, and Immunogenicity of Varying Types of COVID-19 Vaccines in Children Younger Than 18 Years: An Update of Systematic Review and Meta-Analysis

**DOI:** 10.3390/vaccines11010087

**Published:** 2022-12-30

**Authors:** Yan Tian, Long Chen, Yuan Shi

**Affiliations:** 1Department of Neonatology, Children’s Hospital of Chongqing Medical University, Chongqing 400014, China; 2National Clinical Research Center for Child Health and Disorders, Chongqing 400014, China; 3Ministry of Education Key Laboratory of Child Development and Disorders, Chongqing 400014, China; 4China International Science and Technology Cooperation Base of Child Development and Critical Disorders, Chongqing 400014, China; 5Chongqing Key Laboratory of Pediatrics, Chongqing 400014, China

**Keywords:** COVID-19 vaccine, SARS-CoV-2, children, adverse reactions, immunogenicity, efficacy, meta-analysis

## Abstract

Vaccination is one of the most effective measures for children as the epidemic progresses. However, there is a significant research gap in the meta-analysis of the COVID-19 vaccines for children younger than 18 years. This study is a comprehensive review of different COVID-19 vaccines. Published articles were retrieved from PubMed, Embase, and the Cochrane Library. Twelve randomized controlled trials (RCTs) of COVID-19 vaccines were included in the review until 21 October 2022. Most local and systemic adverse reactions were predominantly mild to moderate in severity and disappeared quickly after different types of vaccines. The subunit vaccine had the highest safety. The significant risk was lower in the subunit vaccine group after the initial (RR 1.66, 95% CI 1.26–2.17, *p* = 0.0003) and booster vaccination (RR 1.40, 95% CI 1.02–1.92, *p* = 0.04). Younger children had a more outstanding safety profile in the mRNA and inactivated vaccine groups. The humoral immune response was proportional to the number of doses in the inactivated and the adenovirus vaccine groups, and the strength of immunogenicity was negatively correlated with age in the inactivated vaccine. The mRNA and the subunit vaccines provided satisfactory prevention against COVID-19, especially seven days after the booster dose. However, more research and longer-term follow-up are needed to assess the duration of immune responses, efficacy, and safety.

## 1. Introduction

Since the end of 2019, the novel coronavirus (COVID-19) has become a public health threat to people [1], and the pandemic is having an unprecedented impact on the physical and mental health of people around the world [2]. Compared with adults, the proportion of COVID-19 cases in children and adolescents is lower. Although children with COVID-19 seem to have milder symptoms and may even be completely asymptomatic once infected [3,4,5], children can have severe diseases that result in hospitalization, and approximately one-third of adolescents hospitalized for COVID-19 were admitted to an intensive care unit, and 4.9% received invasive mechanical ventilation [6]. In addition, children infected with COVID-19 can develop serious complications, such as multisystem inflammatory syndrome (MIS-C), a severe but rare condition associated with COVID-19, which is a condition where different body parts become inflamed [7,8].

The consequences of the pandemic on children’s development could be vast, with impacts likely on self-control, social competence, and other cognitive abilities [9]. Growing research informs the heavy psychosocial implications of the COVID-19 pandemic, bringing about mental health problems such as anxiety, depression, stress, and maladaptive behavior [10,11,12,13]. According to UNESCO’s report, rising COVID-19 infection rates led to school closures around the world, but limiting the spread of COVID-19 through school closure may lead to reduce interaction with peers, lessen opportunities for physical exercise, and exacerbate adverse psychosocial health outcomes in children, and they have made little or no progress while learning from home [14,15,16,17]. An estimated 1800 schools have had school closures attributable to COVID-19 outbreaks, and more than 900,000 students have been affected [18]. In addition, it was estimated that approximately 1.5 billion young people worldwide had been forced to stay at home, which negatively influenced their social functioning [19].

Moreover, Omicron spreads more easily among children than the previous variants [20], and unvaccinated individuals provide opportunities for more variants to emerge [21]. Therefore, there is an urgent need to vaccinate children against COVID-19 to protect pediatric age groups from harm. A safe and effective vaccine is critically important for infants and young children. Vaccination is one of the effective measures to fight against COVID-19, which can help to reduce the rate of severe diseases [22]. However, there is insufficient evidence that receiving COVID-19 vaccines reduces child mortality or prevents the further spread of the disease, that younger children are at greater risk of spreading COVID-19, or that herd immunity can be achieved through it. Vaccinating children against COVID-19 can protect their mental health [23]. Vaccination reduces family damage due to parental illness, failing economies, and chronic stress [24]. Acquiring the COVID-19 vaccine could provide direct benefits to childhood education by allowing a safer return to school to secure their continued access to education, and letting parents return to full-time work to make the economy recover [25,26]. Therefore, there is an urgent need to protect children through vaccination.

Whether children and adolescents should be vaccinated against COVID-19 remains controversial. Children and adolescents are unique, and parents usually hesitate to vaccinate their children. The vaccine’s novelty and safety concerns can hinder acceptance in the population [27,28]. Several studies and systematic reviews have been performed to demonstrate the safety, immunogenicity, and efficacy of the COVID-19 vaccine. However, there is a lack of experimental data to confirm the safety, efficacy, and immunogenicity of COVID-19 vaccines in children under three years of age and even in infants, as well as experimental data on the different types of vaccines in children younger than 18 years. Therefore, we aimed to comprehensively synthesize the evidence for the safety, efficacy, and immunogenicity of varying types of COVID-19 vaccines in children younger than 18 years as an update to these previously performed systematic reviews.

## 2. Materials and Methods

The systematic review and meta-analysis were conducted following the Preferred Reporting Items for Systematic Reviews and Meta-Analyses (PRISMA) guidelines, and the protocol was registered on PROSPERO (CRD42022369708).

### 2.1. Search Strategy

A systematic retrieval was performed on three databases (PubMed, Embase, and the Cochrane Library) from inception to 21 October 2022. The key search terms were as follows: infant, child, adolescent, COVID-19 vaccines, SARS-CoV-2, COVID-19, randomized controlled trial, and so on (The search details can be found in Appendix A). The clinical trials registers (Clinical Trials.gov, an ongoing NIH trial registry) was also searched for related articles.

### 2.2. Selection Criteria

The inclusion of studies was based on the following criteria: (1) vaccines administered to children aged < 18 years; (2) RCTs comparing COVID-19 vaccines with other vaccines, placebo, adjuvant; (3) reported measures of safety (local and systemic adverse events), immunogenicity (noninferiority of geometric mean titers (GMTs)) or efficacy (COVID-19 infection). The exclusion criteria for the studies were as follows: (1) non-original studies; review, meta-analysis, systematic review, comments, letters, standards, guidelines, or conference abstracts; (2) non-RCT studies, including cohort studies, case-control studies, single-arm studies, cross-sectional studies; (3) animal models; and (4) outcomes without interest.

### 2.3. Data Extraction

After eliminating duplicates, two reviewers (Tian and Chen) screened titles and abstracts and then used predefined criteria to filter the full text of potentially relevant articles. Two authors independently extracted the following information from each of the included studies as outcome indicators: (1) name of the first author, date of publication, intervention measures (vaccine type, number of doses, adjuvant addition, and adjuvant type, etc.), sample size, intervention details; (2) the incidence of adverse events post-vaccination, including total adverse reactions, local adverse reactions, systemic adverse reactions and any specific adverse reactions, such as injection pain, erythema/redness, fever and so on; (3) humoral immune responses and cellular immune responses, including the seroconversion, geometric mean titers (GMT) after vaccination; and (4) incidence of confirmed COVID-19 after vaccination. In the case of differences, a consensus was reached through discussion or consultation with the third author (Shi). Immunogenicity was expressed through the noninferiority of the immune response. The noninferiority criterion indicated if the lower boundary of the 95% confidence interval for the geometric mean ratio was at least 0.67, with or without the difference in the percentage of participants with a serologic response was −10 percentage points or more. The seroconversion was defined as a geometric mean titer (GMT) increase of at least a fourfold increase from baseline after vaccination. The definition of COVID-19 was according to which participants were diagnosed with COVID-19 and if they were positive for SARS-CoV-2 by RT-PCR and with one or more associated symptoms. We carefully read the included studies’ original text and Appendix A to avoid missing data.

### 2.4. Data Analysis

All data were performed using RevMan 5.4.1 statistical software to pool dichotomous through its internal procedures, even if the number of events is 0 in the observation and/or control group. When *I²* values were > 50%, the random effects model was applied to pool the overall results; otherwise, the fixed effects model was used. This study used the risk ratio (RR), and 95% confidence interval (CI) in the case of dichotomous data (RR > 1 represented a risk effect). The *I²* statistic was used to assess the level of statistical heterogeneity. The RR was determined using the formula RR $=\;\frac{\mathrm{incidence}\;\mathrm{in}\;\mathrm{exposed}}{\mathrm{incidence}\;\mathrm{in}\;\mathrm{unexposed}}=\frac{a/\left(a+b\right)}{c/\left(c+d\right)}$ (details can be found in Table 1). The data were deemed heterogeneous when the *I²* values > 50%. *p* values less than 0.05 were considered, and this difference was statistically significant. If we detected heterogeneity, subgroup analyses were conducted to explore the source of heterogeneity. We performed subgroup analyses according to the number of vaccinations, type of vaccines, age of the recipients, and specific adverse reactions and considered sensitivity analyses by excluding pooled studies one by one. To appraise the methodological quality of the included studies, two reviewers (YT and LC) independently assessed each study’s risk according to the Cochrane collaboration tool for assessing the risk of bias as high, low, or unclear for each item. The funnel plot and Egger test were used to judge the publication bias.
(1)$$\mathrm{RR}=\frac{\mathrm{incidence}\;\mathrm{in}\;\mathrm{exposed}}{\mathrm{incidence}\;\mathrm{in}\;\mathrm{unexposed}}=\frac{a/\left(a+b\right)}{c/\left(c+d\right)}$$

Formula (1) The formula for calculating RR.

**Table 1 vaccines-11-00087-t001:** The four-cell table for calculating RR of RCTs.

	Develop Outcome	Do Not Develop Outcome
Exposed	a	b
Not Exposed	c	d

## 3. Results

### 3.1. Characteristics of Included Studies

As shown in the flow diagram in Figure 1, this study found 2276 research articles using the previously mentioned search terms. After removing duplicates, we screened 1505 records based on title and abstract, of which 1454 were determined to be irrelevant. Fifty-one articles were retrieved for full-text assessment. Finally, 12 articles were included in our analysis: 12 articles for safety, seven for immunogenicity, and five for the efficacy of COVID-19 vaccines. These 12 RCTs included four types of COVID-19 vaccines (mRNA, subunit, inactivated, and adenoviral vector vaccines). All included studies reported COVID-19 vaccines from eight countries and regions; a total of 17,731 participants that received the COVID-19 vaccine and 7444 participants who received a placebo were included in this study ranging in age from six months to 17 years old. Frenck et al. and Walter et al. did not provide the exact number of participants in the vaccine group and placebo group in the safety analysis, so we obtained the available data by calculating the product using the form of totals and percentages. Notably, two RCTs [29,30] received a total of three doses. The characteristics of the included studies are summarized in Table 2. We performed the quality assessment for those included studies, the methodological quality of the included studies was high, and the risk of bias was low. Incomplete data and other biases dominated those bias risks. In three studies, incomplete data due to a lack of reasons for exclusive participants during the experiment, two did not specify the method of concealment allocation, and one included a small number of participants, as shown in detail in Figure 2 and Figure 3. Since there were slight differences in outcome indicators among the included studies, this analysis tested publication bias through seven RCTs. Publication bias was performed by funnel plot and Egger’s test, and the results did not show evidence of publication bias in total, systemic, or local adverse reactions (*p* < 0.05) (Appendix A) but did show in the neutralizing antibody 28 days after dose 2 (*p* < 0.05) (Appendix A).

### 3.2. Safety of the COVID-19 Vaccines

#### 3.2.1. Adverse Reactions to Different Introduction Doses

Results showed that the total, systemic, and local adverse reactions after vaccination, both in the mRNA and the adenovirus vector vaccine group, showed a significantly increased risk, and the risk was higher in the second dose than in the first dose (Appendix A and Table 3). The same outcome was observed in the subunit vaccine in the total adverse reactions, but it should be noted that the risk of local adverse reactions was higher in the first dose (RR 2.93, 95%CI 1.76–4.89, *p* < 0.0001; Appendix A and Table 3) than the second dose (RR 1.99, 95%CI 1.24–3.18, *p* = 0.004; Appendix A and Table 3) in the subunit vaccine group. There was no difference in systemic adverse reactions of the subunit vaccine. Of note, we found that only the risk of local reactions after initial vaccination was statistically significant in the inactivated vaccine group (RR 6.34, 95%CI 1.54–26.10, *p* = 0.01; Appendix A and Table 3). The heterogeneity among the above analyses was considerable, and we subsequently performed subgroup analysis for the specific adverse reactions in different vaccine groups.

In the mRNA vaccine groups, we found that the risk of most specific adverse reactions was higher after the booster dose, such as erythema or redness (RR 7.73, 95%CI 3.76–15.90, *p* < 0.00001; Appendix A, Table 4), swelling or hardness (RR 8.59, 95%CI 4.86–15.19, *p* < 0.00001; Appendix A, Table 4), fever (RR 7.85, 95%CI 2.58–23.91, *p* = 0.0003; Appendix A and Table 4) and chills (RR 4.37, 95%CI 3.14–6.09, *p* < 0.00001; Appendix A and Table 4). However, the risk of headache, arthralgia, nausea or vomiting, and loss of appetite after the initial vaccination and the risk of diarrhea and sleepiness after booster vaccination were of no significant difference in the mRNA vaccine group (Appendix A and Table 4).

In addition, in the inactivated vaccine group, the data showed only the risk of local pain after initial vaccination (RR 21.53, 95%CI 3.00–154.35, *p* = 0.002; Appendix A) and booster vaccination (RR 6.84, 95%CI 1.96–23.90, *p* = 0.003; Appendix A) was significantly higher than in the control group, the risk of other specific adverse reactions was of no significant difference compared with the control group. Similar differences were observed in the subunit vaccine, only the risk of local pain after initial vaccination (RR 2.91, 95%CI 1.74–4.84, *p* < 0.0001; Appendix A) and booster vaccination (RR 1.97, 95%CI 1.23–3.16, *p* = 0.005; Appendix A) was statistically significant. Additionally, the data showed that only the risk of local pain (RR 5.67, 95%CI 1.83–17.55, *p* = 0.003; Appendix A) and fever (RR 7.00 95%CI 1.74–28.21, *p* = 0.006; Appendix A) after initial vaccination was statistically significant in the adenovirus vector vaccine.

After pooling whole available data on specific adverse reactions, the significant risk was higher in all vaccine groups than the control group but relatively lower in the subunit vaccine group, both after initial vaccination (RR 1.66, 95% CI 1.26–2.17, *p* = 0.0003; Table 5) and after booster vaccination (RR 1.40, 95% CI 1.02–1.92, *p* = 0.04; Table 5).

#### 3.2.2. Adverse Reactions to Different Age Groups

We observed high heterogeneity in the mRNA vaccine group when subgroup analysis was performed according to different vaccine types. However, the heterogeneity decreased without the RCT study by Anderson et al. The RCT by Anderson et al. was aimed at younger children aged six months to five years, the other four RCTs were conducted in children and adolescents above five years old. Therefore, we performed a subgroup analysis for specific adverse reactions of mRNA vaccine recipients of different ages. For children aged 12–17 years, the risk of almost specific adverse reactions after vaccination was significantly higher, especially erythema/redness (RR 10.74, 95%CI 2.72–43.37, *p* = 0.0007; Appendix A and Table 6) and swelling/hardness (RR 10.61, 95%CI 4.13–27.28, *p* < 0.00001; Appendix A and Table 6) after the first vaccination and erythema/redness (RR 10.16, 95%CI 2.05–50.29, *p* = 0.005; Appendix A and Table 6), swelling/hardness (RR 10.00, 95%CI 2.11–47.24, *p* = 0.004; Appendix A and Table 6) and fever (RR 15.28, 95%CI 10.11–23.11, *p* < 0.00001; Appendix A and Table 6) after the second vaccination. However, there were no statistical differences in headache (RR 1.35, 95%CI 1.00–1.82, *p* = 0.05; Appendix A and Table 6) and nausea/vomiting (RR 1.78, 95%CI 0.82–3.86, *p* = 0.14; Appendix A and Table 6) after the first vaccination. For younger children aged six months–11 years, the risk of swelling/hardness (RR 4.39, 95%CI 2.24–8.58, *p* < 0.0001; Appendix A and Table 6) after the first vaccination and erythema/redness (RR 6.45, 95%CI 2.90–14.31, *p* < 0.00001; Appendix A and Table 6), swelling/hardness (RR 7.71, 95%CI 4.33–13.72, *p* < 0.00001; Appendix A and Table 6) after the booster vaccination were significantly higher. Subsequently, we compared various adverse reactions to vaccination occurrence in older and younger children following the mRNA vaccine. The data suggest a significantly higher risk of specific adverse responses in children aged 12–15 years versus 5–11 years after the booster vaccination (RR 1.84, 95%CI 1.25–2.72, *p* = 0.002; Appendix A). However, there was no statistical difference after the initial vaccination (RR 1.31, 95%CI 0.94–1.82, *p* = 0.11; Appendix A), indicating again that older children were at greater risk of adverse reactions after vaccination than younger children. Anderson et al. chose the mRNA-1273 vaccine as the intervention for children aged six months to five years, and we decided to directly compare the occurrence of various adverse reactions following mRNA-1273 vaccination in children aged 6–23 months and two to five years. Results showed that the risk of various adverse reactions in participants aged 6–23 months was significantly lower than two to five years both after the initial vaccination (RR 0.74, 95%CI 0.71–0.77, *p* < 0.00001; Appendix A and Table 7) and the booster vaccination (RR 0.80, 95%CI 0.77–0.83, *p* < 0.00001; Appendix A and Table 7). Overall, the risk of various adverse reactions after mRNA vaccination appears to be higher in older children aged 12–17 years than in younger children aged six months–11 years. A similar outcome was observed in children aged 6–23 months and two to five years, indicating again that younger children may have a greater safety profile in the mRNA vaccine.

One RCT [30] did not provide information on adverse reactions after the whole vaccination procedure of the inactivated vaccine in different age groups. So, a subgroup analysis was performed according to the age of the participants with two other RCTs. The data showed that only diarrhea (RR 0.21, 95%CI 0.05–0.93, *p* < 0.05; Appendix A) was statistically significant in children younger than 12 years old. In addition, the risk of overall specific adverse reactions was higher in recipients aged 12–17 years than in 3–12 years (RR 2.05, 95%CI 1.58–2.66, *p* < 0.00001; Appendix A), this was consistent with the subgroup analyses in mRNA vaccines, in which younger children may have a greater safety profile.

Subgroup analysis was conducted in the subunit vaccine, in children older than 12 years, only the risks of erythema/redness and nausea/vomiting were not statistically significant, while in children younger than 12 years, all adverse events were not statistically significant (Appendix A). Additionally, there was no significant difference in different age groups (RR 1.22, 95%CI 0.87–1.71, *p* > 0.05; Appendix A).

Further subgroup analysis could not be performed for the adenovirus vector vaccine due to insufficient data for the different age groups.

#### 3.2.3. Adverse Reactions to Different Dose Groups

In the mRNA and inactivated vaccine groups, participants received inconsistent doses. Also, participants in both the subunit vaccine and adenovirus vaccine groups received the same dose, so subgroup analysis failed to be performed on this basis.

### 3.3. Immunogenicity of the COVID-19 Vaccines

A total of 12 studies on the immunogenicity of COVID-19 vaccines were included in this systematic review article. Seven RCTs met the noninferiority of the immune response (detail in Table 8). In particular, Ali et al. also showed a GMR of 1.09 (95% CI: 0.94–1.26) for RBD-binding ELISA antibodies in adolescents aged 12–17 years relative to young adults aged 16–25. In addition, Thuluva et al. reported the GMT of 1099 in adolescents aged 12–17 years and 1148 in 5–12 years, with a neutralizing antibody GMR of 0.82 in 12–18 years and 0.86 in 5–12 years relative to adults, which meet the noninferiority criterion (i.e., the lower limit of two-sided 95% CI of GMT ratio is ≥0.5 limit set) with subunit vaccine as the intervention compared immune responses 14 days after booster vaccination in vaccinees and adults.

#### 3.3.1. Humoral Immune Responses in Different Doses

Five RCTs provided data on seroconversion, which showed that the seroconversion after inoculation was significant, especially after the third dose (RR 392.95, 95%CI 24.66–6260.89, *p* < 0.0001; Appendix A and Table 9) in inactivated vaccine groups. We found an increase in neutralizing antibodies as the number of doses increased in the inactivated vaccine and the adenovirus vaccine groups (Table 9). In addition, the data showed that the neutralizing antibody was significantly increased in 28 days after dose 2 in the subunit vaccine group. The result of Zhu et al. showed the seroconversion rate of RBD-binding antibodies 28 days after dose2 (RR 101.50, 95%CI 6.44–1600.76, *p* = 0.001, Appendix A and Table 10) was higher than dose 1 (RR 99.48, 95%CI 6.31–1569.12, *p* = 0.001, Appendix A and Table 10) in the adenovirus vector vaccine group, and the seroconversion rate of RBD-binding antibodies reached 100% in 28 days after booster vaccination.

#### 3.3.2. Humoral Immune Responses in Different Ages

Publication bias was performed by funnel plot (Egger’s test, *p* = 0.026). Subgroup analysis was performed because three RCTs provided seroconversions for different age groups at 28 days after vaccination. The data showed a significant humoral immune response to SARS-CoV-2 after receiving vaccination in all age groups, but the response appears to be inversely proportional to age, children aged three to five years (RR 125.90, 95%CI 25.72–616.35, *p* < 0.00001; Appendix A and Table 11) have the most robust immune response of the three age groups at 28 days after the second dose. Similar differences were observed after the third dose; the response appears to be relatively high in children aged three to five years (RR 163.67, 95%CI 10.32–2594.58, *p* = 0.0003; Appendix A and Table 11).

#### 3.3.3. Cellular Immune Responses

Two RCTs also assessed the ability of the COVID-19 vaccines to induce T-cell-mediated immunity among participants. Thuluva et al. showed that Th1 significantly skewed cellular immune response after CORBEVAX™ vaccination. Similarly, in the trial of Zhu et al., the data showed that a specific T-cell response was induced at day 28 after primary vaccination, particularly in Th1 cell responses.

### 3.4. Efficacy of the COVID-19 Vaccines

Among the studies on the efficacy of the COVID-19 vaccines, five RCTs were about the mRNA vaccine, three on the mRNA-1273 vaccine, and two were on the BNT162b2 vaccine, with about 100.0% (95% CI: 28.9%-NE%) efficacy was found in Ali et al., 36.8% (12.5% to 54.0%) of 2–5 years old and 50.6% (21.4% to 68.6%) of 6–23 months in Anderson et al., 88.0% (70.0–95.8%) in Buddy Creech et al., 100% (95% CI: 75.3–100%) in Frenck et al., 90.7% (95% CI: 67.4%–98.3%) in Walter et al., and one RCT was on subunit vaccine, about 79.5% (95% CI, 46.8% to 92.1%) efficacy was demonstrated against the predominant circulating Delta variant, in addition, 82.0% (95% CI, 32.4% to 95.2%) efficacy was found due to the SARS-CoV-2 delta variant. Both mRNA vaccines provided satisfactory prevention against COVID-19, especially seven days after the booster dose (RR 0.08, 95%CI 0.03–0.24, *p* < 0.00001; Appendix A and Table 12). Other RCT studies with inactivated, subunit, or adenovirus vector vaccines as interventions did not evaluate the vaccine efficacy.

## 4. Discussion

As the global epidemic spreads, vaccinating children against COVID-19 has become one of the effective measures to prevent the development of the epidemic, but whether children are vaccinated largely depends on the wishes of parents or guardians. Parents with vaccine-hesitant were less knowledgeable about vaccines, the primary reason for concern is the vaccine safety and efficacy [41,42,43]. The children’s age and current physical condition are other consideration factors for parents on vaccination, and parents are reluctant to vaccinate younger children and those who have been sick recently [44]. Other factors influence parents’ vaccination intention and uptakes, such as parents’ age, education, occupation, previous COVID-19 infection, and vaccination status [45,46]. Recent research shows that a high prevalence of severe COVID-19 was in children with comorbidities, such as obesity, diabetes, heart disease, and chronic lung diseases, and that neonate and premature infants also had a high risk [47]. Therefore, vaccination is vital.

An evaluation of COVID-19 vaccines will eliminate parents’ doubts about vaccines and contribute to children’s physical and mental health and all-around development. The findings of our review provide a comprehensive evidence profile on the safety, immunogenicity, and efficacy of COVID-19 vaccines in children younger than 18 years.

Our results show that the most common adverse reactions included local pain, swelling/hardness, and fever after the initial vaccination, and local pain, erythema/redness, swelling/hardness, and fever after the booster vaccination. Still, most local, and systemic adverse reactions were predominantly mild to moderate in severity and transient. Different from Du et al. [48], our meta-analysis found that the adenovirus vaccine was of the lowest safety, while the subunit vaccine was highest in our analyses of the four COVID-19 vaccines; this may be related to our inclusion of the subunit vaccine. However, the RCT [40] with adenoviral vector vaccine as an intervention was a small-sample study, and future studies are still required. In addition, there were no significant differences in total, systemic and local adverse reactions among different dose groups for various vaccines. Our results indicated that younger children may have a greater safety profile in the mRNA vaccine group and the inactivated vaccine group.

Good immunogenicity was observed in the included vaccine types. We found that the immune response to the mRNA and the subunit vaccine in adolescents was non-inferior in young people, consistent with the previous systemic review [49]. It was found that the humoral immune response is proportional to the number of doses in the inactivated and the adenovirus vaccine groups in our meta-analysis. In addition, our analysis found dose-level-dependent immunogenicity in the inactivated vaccine, which was in line with a newly published meta-analysis conducted by Du et al. [48]. The data of Han et al. showed that the higher dose of the vaccine could induce stronger immune responses in all age groups compared with the lower dose of the adenovirus vector vaccine. However, there are some different findings in our analysis; the immunogenicity’s strength was negatively correlated with age in the inactivated vaccine. Some possibilities have been suggested. Other vaccines given to children produce a strong immune response to provide a better immune environment and generate cross-reactivity among the different beta coronaviruses, which may confer a nonspecific protective effect against SARS-CoV-2, such as measles, mumps, and rubella [50,51,52]. The immunity responses decreased with aging, indicating that a booster vaccine may be needed. However, the data showed a lower seroconversion rate and neutralizing antibody titer on day 28 in younger children (three to five years) than that of other age cohorts by Xia et al. [30]; future studies are still required to explore this result. The mRNA vaccines and subunit vaccines also elicited robust binding antibody responses to the prototype SARS-CoV-2, as well as against more recent variants: Alpha, Beta, Delta, and Omicron, including subvariants BA.1, BA.2, and BA.5 and the B.1.351 (beta), B.1.617.2 (delta), and Omicron variants. The data from Thuluva et al. showed the cellular immune response in the pediatric population demonstrated the expected Th1 skew. However, the specific T-cell response was not enhanced after booster vaccination by Zhu et al. [40].

Our analysis showed that the mRNA and subunit vaccines provided satisfactory protection against prototype SARS-CoV-2 and more recent variants. It should be especially noted that the research of Anderson et al. showed a lower efficacy, in which B.1.1.529 (Omicron) was the predominant circulating variant at the time of this experiment. The effectiveness of the mRNA vaccine declined during the Omicron period, and a similar phenomenon was also observed in other research on children and adults [53,54,55]. Like the previous studies, the vaccines still have a protective effect even during an epidemic of a new variant [56,57,58], which also led to significant heterogeneity in effectiveness analyses, and there was little change in the meta-analysis result with Anderson’s RCT removed, indicating that the analysis results were robust.

Compared with the previous meta-analysis, this is the first meta-analysis to include children aged six months to three years old on COVID-19 vaccines. In addition, a new type of vaccine (subunit vaccine) has been added to our analysis, providing a more comprehensive assessment of existing vaccines’ safety, immunogenicity, and efficacy. Additionally, our review evaluates the effectiveness of the mRNA vaccine against the Omicron variants. This review included the latest high-quality randomized controlled studies and had a large sample size, with 17,731 participants in the experimental group and 7444 in the control group, which provides strong evidence for vaccine evaluation.

There are several limitations in our systematic review and meta-analysis. First, there is a lack of data on younger children under six months and a lack of longer-term follow-up to assess the duration of immune responses, efficacy, and safety for children younger than 18. In addition, our analysis included four types of COVID-19 vaccines (the mRNA, inactivated, subunit, and adenovirus vector vaccine); however, there was just one RCT about the adenoviral vector vaccine as an intervention with a small sample, and only two RCTs provided relevant data on cellular immune responses in our analysis. Further studies are still required. In addition, our meta-analysis did not evaluate the COVID-19 vaccines in high-risk children, nor did we evaluate the effectiveness of the COVID-19 vaccine by hospitalization, severe illness, and mortality rates of children in the vaccine and control groups, due to limited data. Last and most importantly, high unexplained heterogeneity could be found in some subgroups in our review, which might be attributed to the variation in different variants, the design of studies, vaccine dose, sociodemographic factors, etc. Therefore, the safety, immunogenicity, and efficacy of different COVID-19 vaccines in children younger than 18 years, especially under six months, still require extensive and high-quality studies and longer follow-up periods.

The following questions remain about the vaccination of children under the age of 18. First, long-term follow-up is needed to assess the duration and efficacy of the immune response to COVID-19 vaccines. Second, it is urgent to evaluate whether the COVID-19 vaccines cause severe side effects such as glomerulonephritis, myocarditis, and chronic fatigue syndrome. Finally, more attention should be given to vaccinating high-risk children to protect them from contracting COVID-19.

## 5. Conclusions

Based on the systematic analysis of the four COVID-19 vaccines, we found that the four vaccines are generally safe and feasible with no serious side effects, but considering that some vaccines have been less studied, further research is needed. The immunogenicity and effectiveness of the four vaccines in children younger than 18 years are acceptable and approved, which may improve parents’ confidence in COVID-19 vaccinations. However, there are no data on children younger than six months, and more research are needed. Longer-term follow-up is required to assess the duration of immune responses, efficacy, and safety.

## Figures and Tables

**Figure 1 vaccines-11-00087-f001:**
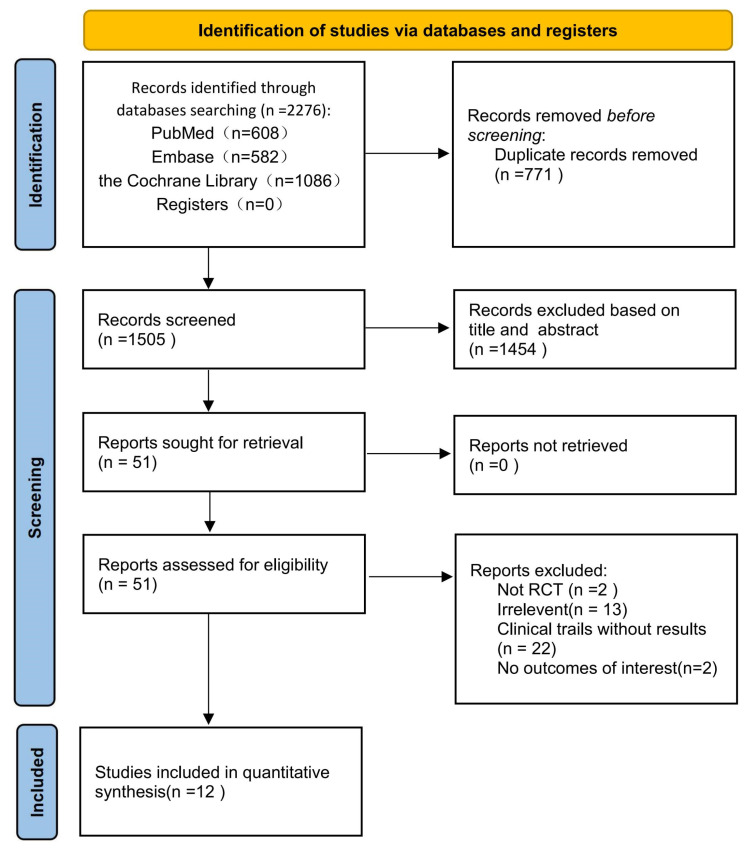
Flow chart of study identification and selection.

**Figure 2 vaccines-11-00087-f002:**
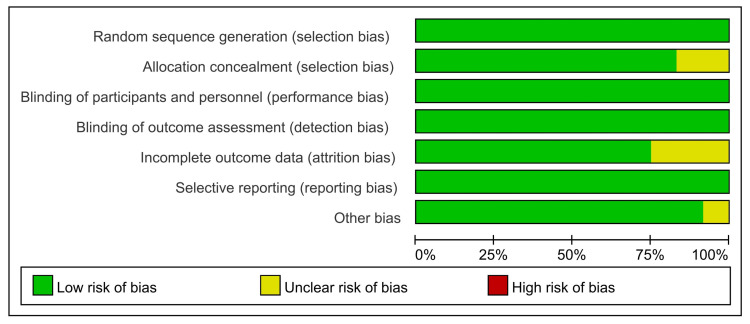
Risk of bias graph for included RCTs.

**Figure 3 vaccines-11-00087-f003:**
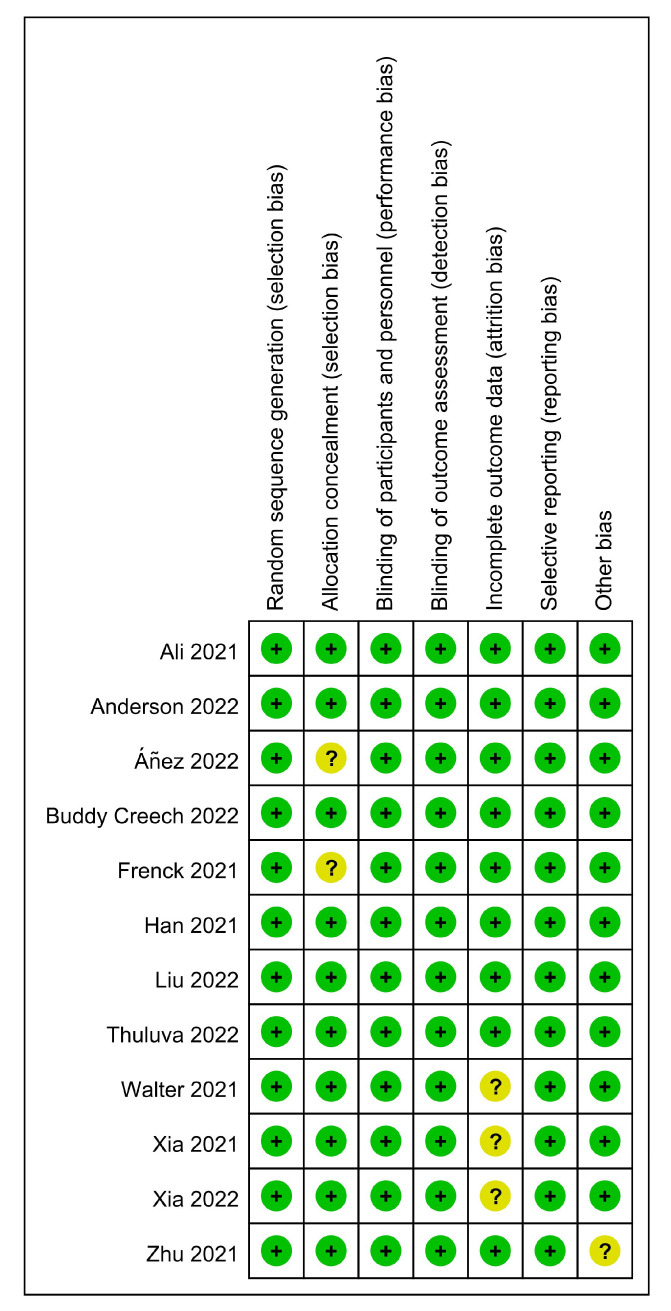
Risk of bias summary for included RCTs (the green color and special symbol “+” are represented a low risk of bias, and the yellow color and special symbol “?” are represented an unclear risk of bias).

**Table 2 vaccines-11-00087-t002:** The characteristics of the included studies.

Author, Year	Country	Phase	Age (Years)	Type of Vaccine	Dose of Administration (Per Dose)	Time of Inoculations (Days)	Control	No. of the Observation Group	No. of the Control Group
Ali et al. [31]	The USA	phase 2–3	12–17	mRNA-1273 vaccine (mRNA vaccine)	100 μg	0, 28	Saline	2486	1240
Anderson et al. [32]	the USA, Canada	phase 2–3	6 Months-5	mRNA-1273 vaccine (mRNA vaccine)	25 μg, 50 μg	0, 28	Saline	5011	1751
Áñez et al. [33]	the USA, Mexico	phase 3	12–17	NVX-CoV2373 (subunit vaccine)	0.5 mL	0, 21	Saline	1487	745
Buddy Creech et al. [34]	the USA, Canada	phase 2–3	6–11	mRNA-1273 vaccine (mRNA vaccine)	50 μg	0, 28	Saline	3385	995
Frenck et al. [35]	The USA	phase 3	12–15	BNT162b2 Vaccine (mRNA vaccine)	30 μg	0, 21	Saline	1131	1129
Han et al. [36]	China	phase 1–2	3–17	CoronaVac (Inactivated vaccine)	1.5 μg, 3.0 μg	0, 28	Aluminum hydroxide adjuvant	436	114
Liu et al. [37]	China	phase 2	12–17	MVC-COV1901 (subunit vaccine)	0.5 mL	0, 28	Saline	341	58
Thuluva et al. [38]	India	phase 2–3	5–17 (<12- ≥5, <18- ≥12)	CORBEVAX™ (subunit vaccine)	0.5 mL	0, 28	Placebo (Not noted)	468	156
Walter et al. [39]	the USA, Spain, Finland, Poland	phase 2–3	5–11	BNT162b2 Vaccine (mRNA vaccine)	10 μg	0, 21	Saline	1518	750
Xia et al. [29]	China	phase 1–2	3–17 (3–5, 6–12, 13–17)	WBIBP-CorV (Inactivated vaccine)	2.5 μg, 5 μg, 10 μg	0, 28, 56	Alum	612	204
Xia et al. [30]	China	phase 1–2	3–17 (3–5, 6–12, 13–17)	BBIBP-COV (Inactivated vaccine)	2 μg,4 μg, 8 μg	0, 28, 56	Saline and aluminum hydroxide adjuvant	756	252
Zhu et al. [40]	China	phase 2	6–17	Ad5-vectored COVID-19 vaccine (Adenovirus vaccine)	0.3 mL	0, 56	Placebo containing the same excipients as the vaccine, without viral particles	100	50

**Table 3 vaccines-11-00087-t003:** All adverse reactions in the vaccination group versus the control group.

		No. of Studies	RR (95% CI)	I²	*p*-Value
**Total Adverse Reactions**					
mRNA vaccine	After dose 1	3	1.30 [1.07, 1.57]	98%	<0.05
	After dose 2	3	1.43 [1.14, 1.79]	98%	<0.05
Inactivated vaccine	After dose 1	1	1.27 [0.76, 2.13]	Not applicable	>0.05
	After dose 2	1	1.83 [0.90, 3.72]	Not applicable	>0.05
Submit vaccine	After dose 1	1	1.57 [1.17, 2.11]	Not applicable	<0.05
	After dose 2	1	1.94 [1.26, 2.98]	Not applicable	<0.05
Adenovirus vector vaccine	After dose 1	1	3.44 [1.78, 6.65]	Not applicable	<0.05
	After dose 2	1	8.25 [2.06, 33.00]	Not applicable	<0.05
**Systemic adverse reactions**					
mRNA vaccine	After dose 1	3	1.13 [1.02, 1.24]	88%	<0.05
	After dose 2	3	1.47 [1.08, 2.01]	99%	<0.05
Inactivated vaccine	After dose 1	1	1.32 [0.87, 2.00]	Not applicable	>0.05
	After dose 2	1	1.61 [0.76, 3.40]	Not applicable	>0.05
Submit vaccine	After dose 1	1	1.11 [0.78, 1.57]	Not applicable	>0.05
	After dose 2	1	1.22 [0.72, 2.09]	Not applicable	>0.05
Adenovirus vector vaccine	After dose 1	1	3.70 [1.55, 8.83]	Not applicable	<0.05
	After dose 2	1	6.00 [1.48, 24.38]	Not applicable	<0.05
**Local adverse reactions**					
mRNA vaccine	After dose 1	3	1.80 [1.11, 2.92]	99%	<0.05
	After dose 2	3	1.93 [1.25, 2.97]	99%	<0.05
Inactivated vaccine	After dose 1	1	6.34 [1.54, 26.10]	Not applicable	<0.05
	After dose 2	1	4.29 [1.03, 17.96]	Not applicable	=0.05
Submit vaccine	After dose 1	1	2.93 [1.76, 4.89]	Not applicable	<0.05
	After dose 2	1	1.99 [1.24, 3.18]	Not applicable	<0.05
Adenovirus vector vaccine	After dose 1	1	6.00 [1.94, 18.53]	Not applicable	<0.05
	After dose 2	1	19.69 [1.21,319.62]	Not applicable	<0.05

Note: RR, Risk ratio; CI, confidence interval; *p* < 0.05.

**Table 4 vaccines-11-00087-t004:** Specific adverse reactions in the mRNA vaccine group versus the control group after dose 1 and dose 2.

		After Dose 1			After Dose 2		
	No. of Studies	RR (95% CI)	I²	*p*-Value	RR (95% CI)	I²	*p*-Value
**Overall**	5	1.91 [1.70, 2.16]	97	<0.05	3.13 [2.73, 3.59]	97	<0.05
**Local pain**	5	2.32 [1.72, 3.13]	98	<0.05	2.54 [1.89, 3.42]	98	<0.05
**Erythema or Redness**	5	5.66 [2.75, 11.65]	92	<0.05	7.73 [3.76, 15.90]	92	<0.05
**Swelling or Hardness**	5	6.21 [3.14, 12.28]	90	<0.05	8.59 [4.86, 15.19]	84	<0.05
**Axillary Swelling**	3	1.85 [1.15, 2.98]	93	<0.05	2.90 [2.02, 4.18]	84	<0.05
**Fever**	5	3.31 [1.47, 7.45]	92	<0.05	7.85 [2.58, 23.91]	96	<0.05
**Headache**	4	1.14 [0.92, 1.43]	94	>0.05	2.04 [1.63, 2.56]	94	<0.05
**Fatigue**	4	1.29 [1.16, 1.43]	79	<0.05	2.08 [1.70, 2.54]	93	<0.05
**Myalgia**	4	1.59 [1.39, 1.81]	43	<0.05	2.87 [2.07, 3.98]	90	<0.05
**Arthralgia**	4	1.10 [0.84, 1.45]	77	>0.05	2.22 [1.50, 3.28]	89	<0.05
**Nausea or Vomiting**	4	1.41 [0.99, 1.99]	75	=0.05	2.55 [2.23, 2.92]	0	<0.05
**Chills**	4	1.63 [1.15, 2.33]	89	<0.05	4.37 [3.14, 6.09]	86	<0.05
**Diarrhea**	2	1.27 [0.96, 1.67]	23	>0.05	1.21 [0.82, 1.80]	55	>0.05
**Irritability or Crying**	1	1.08 [1.01, 1.16]	Not applicable	<0.05	1.10 [1.01, 1.19]	Not applicable	<0.05
**Sleepiness**	1	0.97 [0.86, 1.09]	Not applicable	>0.05	1.04 [0.90, 1.19]	Not applicable	>0.05
**Loss of appetite**	1	1.12 [0.96, 1.30]	Not applicable	>0.05	1.25 [1.06, 1.48]	Not applicable	<0.05

Note: RR, Risk ratio; CI, confidence interval; *p* < 0.05.

**Table 5 vaccines-11-00087-t005:** Overall specific adverse reactions among the vaccination group versus the control group after dose 1 and dose 2.

		After Dose 1			After Dose 2		
	No. of Studies	RR (95% CI)	I²	*p*-Value	RR (95% CI)	I²	*p*-Value
mRNA vaccine	5	1.91 [1.70, 2.16]	97	<0.05	3.13 [2.73, 3.59]	97	<0.05
Inactivated vaccine	2	1.76 [1.20, 2.57]	38	<0.05	2.18 [1.30, 3.67]	0	<0.05
Subunit vaccine	1	1.66 [1.26, 2.17]	19	<0.05	1.40 [1.02, 1.92]	12	<0.05
Vectored vaccine	1	5.27 [2.80, 9.91]	0	<0.05	6.21 [2.40, 16.11]	0	<0.05

Note: RR, Risk ratio; CI, confidence interval; *p* < 0.05.

**Table 6 vaccines-11-00087-t006:** Specific adverse reactions in mRNA vaccine recipients of different ages.

		≥12 Years				<12 Years			
		No. of Studies	RR (95% CI)	I²	*p*-Value	No. of Studies	RR (95% CI)	I²	*p*-Value
**After dose 1**	**Local pain**	2	3.15 [2.27, 4.37]	96	<0.05	3	1.89 [1.44, 2.48]	97	<0.05
	**Erythema or Redness**	2	10.74 [2.72, 43.37]	89	< 0.05	3	3.78 [1.96, 7.29]	89	<0.05
	**Swelling or Hardness**	2	10.61 [4.13, 27.28]	81	<0.05	3	4.39 [2.24, 8.58]	85	<0.05
	**Fever**	2	5.00 [1.40, 17.82]	89	<0.05	3	2.50 [1.02, 6.17]	90	=0.05
	**Headache**	2	1.35 [1.00, 1.82]	96	=0.05	2	0.98 [0.89, 1.07]	0	>0.05
	**Fatigue**	2	1.38 [1.24, 1.55]	71	<0.05	2	1.19 [1.03, 1.38]	72	<0.05
	**Myalgia**	2	1.70 [1.52, 1.90]	14	<0.05	2	1.40 [1.18, 1.67]	0	<0.05
	**Arthralgia**	2	1.33 [1.15, 1.55]	0	<0.05	2	0.84 [0.45, 1.56]	85	>0.05
	**Nausea or Vomiting**	2	1.78 [0.82, 3.86]	81	>0.05	2	1.24 [0.67, 2.27]	62	>0.05
	**Chills**	2	2.08 [1.31, 3.30]	92	<0.05	2	1.25 [0.83, 1.87]	69	>0.05
**After dose 2**	**Local pain**	2	3.64 [2.55, 5.19]	95	<0.05	3	1.99 [1.70, 2.34]	92	<0.05
	**Erythema or Redness**	2	10.16 [2.05, 50.29]	93	<0.05	3	6.45 [2.90, 14.31]	91	<0.05
	**Swelling or Hardness**	2	10.00 [2.11, 47.24]	93	<0.05	3	7.71 [4.33, 13.72]	78	<0.05
	**Fever**	2	15.28 [10.11, 23.11	4	<0.05	3	5.07 [1.14, 22.44]	97	<0.05
	**Headache**	2	2.50 [2.14, 2.91]	79	<0.05	2	1.66 [1.35, 2.04]	77	<0.05
	**Fatigue**	2	2.47 [2.20, 2.78]	62	<0.05	2	1.75 [1.55, 1.98]	55	<0.05
	**Myalgia**	2	3.80 [3.35, 4.31]	0	<0.05	2	2.11 [1.41, 3.16]	81	<0.05
	**Arthralgia**	2	3.14 [2.68, 3.66]	0	<0.05	2	1.57 [1.11, 2.22]	57	<0.05
	**Nausea or Vomiting**	2	2.78 [2.31, 3.36]	0	<0.05	2	2.33 [1.92, 2.82]	0	<0.05
	**Chills**	2	5.85 [5.03, 6.79]	0	<0.05	2	4.37 [3.14, 6.09]	72	<0.05

Note: RR, Risk ratio; CI, confidence interval; *p* < 0.05.

**Table 7 vaccines-11-00087-t007:** Specific adverse reactions in mRNA vaccine recipients aged 6–23 months versus two to five years.

		After Dose 1			After Dose 2		
	No. of Studies	RR (95% CI)	I²	*p*-Value	RR (95% CI)	I²	*p*-Value
**Overall**	1	0.74 [0.71, 0.77]	97	<0.05	0.80 [0.77, 0.83]	98	<0.05
**Any local adverse reactions**	1	0.70 [0.66, 0.74]	Not applicable	<0.05	0.73 [0.70, 0.77]	Not applicable	<0.05
**Local pain**	1	0.60 [0.57, 0.64]	Not applicable	<0.05	0.64 [0.60, 0.67]	Not applicable	<0.05
**Erythema or Redness**	1	1.53 [1.24, 1.89]	Not applicable	<0.05	1.53 [1.29, 1.80]	Not applicable	<0.05
**Swelling or Hardness**	1	1.85 [1.49, 2.31]	Not applicable	<0.05	1.83 [1.55, 2.15]	Not applicable	<0.05
**Axillary swelling**	1	0.91 [0.73, 1.13]	Not applicable	>0.05	1.02 [0.85, 1.23]	Not applicable	>0.05

Note: RR, Risk ratio; CI, confidence interval; *p* < 0.05.

**Table 8 vaccines-11-00087-t008:** Immunogenicity of included studies.

Included Studies	Type of Vaccine	Days	Age (Years)	Dose	Participants	GMT (IU/mL)	GMR	Serologic Response	The Difference in Serologic Response	Noninferiority
Ali et al. [31]	mRNA-1273 vaccine (mRNA vaccine)	57	12–17	100 µg	340	1401.7 (1276.3, 1539.5)	1.08 (0.94, 1.25)	336/340 (97.0, 99.7)	0.2 (−1.8, 2.4)	Yes
			18–25	NA	296	1301.3 (1177.0, 1438.8)		292/296 (96.6, 99.6)		
Anderson et al. [32]	mRNA-1273 vaccine (mRNA vaccine)	57	6–23 months	25 μg	230	1781 (1616, 1962)	1.3 (1.1, 1.5)	230/230 (98.4, 100.0)	0.7 (−1.0, 2.5)	Yes
			2–5	25 μg	264	1410 (1272, 1563)	1.0 (0.9, 1.2)	261/264 (96.7, 99.8)	−0.4 (−2.7, 1.5)	Yes
			18–25	100 μg	294	1391 (1263, 1531)	/	289/291 (97.5, 99.9)	/	/
Áñez et al. [33]	NVX-CoV2373 (subunit vaccine)	35	12–17	0.5 mL	390	3860 (3423, 4352)	1.5 (1.3, 1.7)	-/390 (98.7%) (97.0, 99.6)	−1.0 (−2.8, 0.2)	Yes
			18–25	NA	416	2634 (2398, 2904)		-/416 (99.8%) (98.7, 100)		
Buddy Creech et al. [34]	mRNA-1273 vaccine (mRNA vaccine)	57	6–11	50 μg	320	1610.2 (1456.6, 1780.0)	1.2 (1.1, 1.4)	313/316 (97.3, 99.8)	0.1 (−1.9, 2.1)	Yes
			18–25	100 μg	295	1299.9 (1170.6, 1443.4)		292/295 (97.1, 99.8)		
Frenck et al. [35]	BNT162b2 Vaccine (mRNA vaccine)	A month after dose 2	12–15	30 µg	190	1239.5 (1095.5, 1402.5)	1.76 (1.47, 2.10)	NA	NA	Yes
			16–25	NA	170	705.1 (621.4, 800.2)		NA		
Liu et al. [37]	MVC-COV1901 (subunit vaccine)	57	12–17	0.5 mL	334	648.47 (608.62, 690.93)	1.16 (1.04, 1.29)	334/334 (98.90, 100.00)	−0.0% (0.00, 0.00)	Yes
			20–30	NA	210	559.54 (512.05, 611.34)		210/210 (98.26, 100.00)		
Walter et al. [39]	BNT162b2 vaccine (mRNA vaccine)	A month after dose 2	5–11	10 μg	264	1197.6 (1106.1, 1296.6)	1.04 (0.93, 1.18)	NA	NA	Yes
			16–25	30 μg	253	1146.5 (1045.5, 1257.2)		NA		

Note: GMT, geometric mean titers; GMR, geometric mean ratio; NA, not available.

**Table 9 vaccines-11-00087-t009:** Neutralizing antibody in the vaccine groups versus the control groups.

Vaccine Type	Time	No. of Studies	RR (95% CI)	I²	*p*-Value
**Inactivated vaccine**	28 days after dose 1	2	245.69 [34.93, 1727.83]	67	<0.05
	28 days after dose 2	3	363.09 [73.82, 1785.92]	0	<0.05
	28 days after dose 3	1	392.95 [24.66, 6260.89]	Not applicable	<0.05
**Subunit vaccine**	28 days after dose 2	1	31.29 [6.48, 151.07]	Not applicable	<0.05
**Adenovirus vaccine**	28 days after dose 1	1	14.67 [4.88, 44.04]	Not applicable	<0.05
	28 days after dose 2	1	24.50 [6.30, 95.28]	Not applicable	<0.05

Note: RR, Risk ratio; CI, confidence interval; *p* < 0.05.

**Table 10 vaccines-11-00087-t010:** RBD-binding enzyme immunosorbent assay antibody in the adenovirus vaccine group.

	No. of Studies	RR (95% CI)	I²	*p*-Value
28 days after Dose 1	1	99.48 [6.31, 1569.12]	Not applicable	<0.05
28 days after Dose 2	1	101.50 [6.44, 1600.76]	Not applicable	<0.05

Note: RR, Risk ratio; CI, confidence interval; *p* < 0.05.

**Table 11 vaccines-11-00087-t011:** Neutralizing antibody in the inactivated vaccine groups versus the control groups.

	No. of Studies	RR (95% CI)	I²	*p*-Value
**Neutralizing antibody 28 days after Dose 2**				
3–5 years old	3	125.90 [25.72, 616.35]	0	<0.05
6–11/12 years old	3	122.82 [25.05, 602.16]	0	<0.05
12/13–17 years old	3	117.87 [24.04, 577.88]	0	<0.05
**Neutralizing antibody 28 days after Dose 3**				
3–5 years old	1	163.67 [10.32, 2594.58]	Not applicable	<0.05
6–12 years old	1	120.30 [7.61, 1901.57]	Not applicable	<0.05
13–17 years old	1	112.80 [7.13, 1783.53]	Not applicable	<0.05

Note: RR, Risk ratio; CI, confidence interval; *p* < 0.05.

**Table 12 vaccines-11-00087-t012:** COVID-19 was diagnosed after vaccination in the vaccine group versus the control group.

	No. of Studies	RR (95% CI)	I²	*p*-Value
**COVID-19 after the vaccination**				
After dose 1 to before dose 2	2	0.16 [0.08, 0.32]	0	<0.05
Within 7 days after dose 2	1	0.09 [0.01, 1.64]	Not applicable	>0.05
7 days after dose 2	2	0.08 [0.03, 0.24]	0	<0.05
14 days after dose 2	3	0.30 [0.09, 0.97]	62	<0.05
**COVID-19 after dose 2**				
mRNA-1273 vaccine	3	0.30 [0.09, 0.97]	62	<0.05
BNT162b2 COVID-19 Vaccine	2	0.08 [0.03, 0.24]	0	<0.05

Note: RR, Risk ratio; CI, confidence interval; *p* < 0.05.

## Data Availability

All data are included and/or available within the Appendix A. Additional data about this paper may be requested from the authors.

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
