# Peer review of "Safety, Efficacy, and Immunogenicity of Varying Types of COVID-19 Vaccines in Children Younger Than 18 Years: An Update of Systematic Review and Meta-Analysis"

_vaccines, 2022, doi:10.3390/vaccines11010087_

Round 1

Reviewer 1 Report

Analysis of adverse reactions in children from multiple types of vaccines will provide important insights not only for countermeasures against novel coronavirus infection, but also for future vaccine development.

L123- In the Data analysis section, it would be easier to understand if the formula for calculating RR is shown; it is difficult to understand if there is a difference in the method of calculating RR between the M-H fixed case and the M-H random case. It is also necessary to describe how the group with 0 events is treated when calculating the RR (for example, for seroconversion in Figure S17, the number of events is 0 for the control group because seroconversion does not occur, but how is the RR calculated? (e.g., how is the RR calculated for seroconversion in Figure S17?)

L352-.

How do you verify the safety of the vaccine in high-risk children?

Do we expect indirect protection of the high-risk group from infectious diseases through the intake of vaccines by those around them, or direct protection through vaccination of the high-risk group themselves?

Also, in COVID-19, the effectiveness of the vaccine should be demonstrated by comparing the hospitalization rate, serious illness rate and mortality rate of children in the vaccine and control groups, and the difference in the number of diagnoses should not directly indicate safety. Considering the aspect of high number of adverse reactions, it may be difficult to conclude that the vaccine is safe.

In the main body of the review, RR=Relative risk, but in the supplement figure, RR=Risk ratio. This needs to be unified.

Regarding Age in Ali et al (31), which is used for analysis in the review, Table 1 lists 12-17, but Table 8 lists 18-25 in addition to 12-17. For the other studies, there are also differences in Age groups between Table 1 and 8.

Figure S18, Table 10, etc.

I am not sure how to calculate Risk ratio.Figure S18 A) In the "total" column of COVID-19 after the vaccination, the number of events in the vaccine group is 187/16261, while the number of events in the control group is 175/8177. In this case, the RR would be about 0.54, but in fact it is shown as 0.17.In other tables, the RR values seem to differ from those calculated from the actual number of events, so it is necessary to show how the RR is calculated.

Author Response

Dear Reviewer,

Thank you very much for your constructive comments and suggestions on our manuscript (Manuscript ID: vaccines-2100003). We have changed the name of the manuscript to “Safety, Efficacy, and Immunogenicity of Varying Types of COVID-19 Vaccines in Children Younger Than 18 Years: An Update of Systematic Review and Meta-Analysis”. We have revised the manuscript accordingly, and all amendments are indicated by the red font in the revised manuscript (Please see the attachment). In addition, our point-by-point responses to the comments are listed below this letter.

We hope that our revised manuscript has been substantially improved and is now acceptable for publication in your journal and look forward to hearing from you soon.

With best wishes,

Yours sincerely,

Yuan Shi

Responses to Reviewer’ s Comments

  1. In the Data analysis section, it would be easier to understand if the formula for calculating RR is shown.

Response: We thank the reviewer for pointing out this issue in our previous manuscript. We have added the formula to calculate RR in our revised manuscript (L142-146).

  1. It is difficult to understand if there is a difference in the method of calculating RR between the M-H fixed case and the M-H random case.

Response: Thank you for your thoughtful advice. According to our learning, the statistical methods of those two cases in the statistical software Revman are slightly different, but the purpose is to make the results of the meta-analysis more credible and more accurate to represent the actual effect. Therefore, When values were > 50%, the random effects model was applied to pool the overall results; otherwise, the fixed effects model was used to compute.

  1. It is necessary to describe how the group with 0 events is treated when calculating the RR.

Response: Thank you for your helpful advice. All data were performed using RevMan 5.4.1 statistical software to pool dichotomous through its internal procedures, even though the number of events is 0 in the observation and/or control group. We have described how to calculate RR when the number of events is 0 in our revised manuscript (L127-129).

  1. How do you verify the safety of the vaccine in high-risk children?

Response: We thank the reviewer for the thoughtful comment. The included 12 RCTs incorporate healthy children as well as participants with chronic diseases (such as asthma, diabetes, hepatitis B, and human immunodeficiency virus infections). But unfortunately, those RCTs did not do a separate analysis of participants with underlying medical conditions, so we cannot analyze the safety of COVID-19 vaccines in high-risk children. We have included this limit in our revised manuscript (L437-439).

  1. Do we expect indirect protection of the high-risk group from infectious diseases through the intake of vaccines by those around them, or direct protection through vaccination of the high-risk group themselves?

Response: Thank you for pointing out this issue in our manuscript. We believe that in the early stage of COVID-19 vaccination, people around high-risk groups can be vaccinated to protect themselves. The safety, effectiveness, and immunogenicity of the vaccine can be evaluated by following up with the vaccinated people for a period, and then according to the evaluation results to judge whether high-risk groups can be vaccinated. If high-risk groups can be vaccinated, they can be protected by vaccinating themselves.

  1. In COVID-19, the effectiveness of the vaccine should be demonstrated by comparing the hospitalization rate, serious illness rate, and mortality rate of children in the vaccine and control groups.

Response: Thank you for your valuable comments, we agree with your suggestions. But unfortunately, no data was comparing the rates of hospitalization, serious illness, and mortality of children in the COVID-19 vaccine groups and the control groups in the included studies, and only the number of people diagnosed with COVID-19 after vaccination were described. We have included this limit in our revised manuscript (L437-439). We will perform a meta-analysis based on your suggestions to evaluate the effectiveness of COVID-19 vaccines in the future.

  1. In the main body of the review, RR=Relative risk, but in the supplement figure, RR=Risk ratio. This needs to be unified

Response: We thank the reviewer’s careful reminders, and we apologize for our careless mistake sincerely. We have replaced RR=Relative risk with RR=Risk ratio in the main body of the review which is indicated by the red font in our revised manuscript.

  1. Considering the aspect of high number of adverse reactions, it may be difficult to conclude that the vaccine is safe.

Response: We thank the reviewer for the thoughtful comment. Participants showed a wide variety of adverse reactions after receiving the vaccine, which is nearly consistent with those seen with vaccines against other infectious diseases. In addition, after in-depth reading of the included literature, we found that these adverse reactions were mild and short-lived without serious side effects, which could be accepted by most people. Therefore, we think that the safety of the COVID-19 vaccines is acceptable.

  1. Regarding Age in Ali et al (31), which is used for analysis in the review, Table 1 lists 12-17, but Table 8 lists 18-25 in addition to 12-17. For the other studies, there are also differences in Age groups between Table 1 and 8.

Response: We thank the reviewer for pointing out this issue. Table 8 is the immunogenicity of the included studies. The immunogenicity was expressed through the noninferiority of the immune response. The noninferiority was reflected by comparing the geometric mean ratio (GMT) and the serologic responses in children and young adults (such as 18-25 and other ages). Notably, the data (as 18-25 and others) of immunogenicity in young adult participants were provided from previous experiments aimed at adults.

  1. I am not sure how to calculate the Risk ratio. Figure S18 A) In the "total" column of COVID-19 after the vaccination, the number of events in the vaccine group is 187/16261, while the number of events in the control group is 175/8177. In this case, the RR would be about 0.54, but in fact it is shown as 0.17. In other tables, the RR values seem to differ from those calculated from the actual number of events, so it is necessary to show how the RR is calculated.

Response: We thank the reviewer for pointing out this issue in our manuscript. The RR is calculated by the RevMan 5.4.1 statistical software according to the provided formula, but the RR in the "total" column is calculated by pooling the RRs by the software using its unique algorithm, it is inconsistent with RR calculations that are not pooled.

Reviewer 2 Report

This is a review of 12 publications about COVID-19 vaccination adverse events published in 2021 and 2022. 5 studies from China and so far I can prove non of the studies are done during the omicron wave.

Unfortunately, the authors repeat in the introduction all the wrong predictions of the experts from the early phase of the pandemic, which led to wrong political decisions to the detriment of the children:

-        There is no excess mortality from COVID-19 in children

-        Younger children were not at greater risk of spreading COVID-19

-        There is no evidence that vaccination reduces mortality in children or prevents the further spread of the disease

-        There is no indication to realize herd immunity by vaccination

-        At this point, I stop to read this paper and wrote the review above

The strength of the paper is that the authors use the current recognized methods for a review.

However, nobody is interested in local reactions, fever, or headaches for a short time. That’s what my patients expect and I never have recalls because of local reactions.
The problem is the patients who develop Glomerulonephritis, Myocarditis, or chronic fatigue syndrome for a long time. I find no data about this despite having many patients with these post-VAC problems for more than a year in my praxis.

The immunogenicity data are known from registration studies and are plausible.

In the introduction, the authors have to declare that many of our arguments to vaccinate children against COVID-19 (see point 1) have not come true from today’s perspective

-        By recognizing this, long-term, disabling side effects like Glomerulonephritis, Myocarditis, or chronic fatigue syndrome are of high importance

The question is why these important side effects are not recognized in so many expensive studies

Author Response

Dear Reviewer,

Thank you very much for your constructive comments and suggestions on our manuscript (Manuscript ID: vaccines-2100003). We have changed the name of the manuscript to “Safety, Efficacy, and Immunogenicity of Varying Types of COVID-19 Vaccines in Children Younger Than 18 Years: An Update of Systematic Review and Meta-Analysis”. We have revised the manuscript accordingly, and all amendments are indicated by the red font in the revised manuscript (Please see the attachment). In addition, our point-by-point responses to the comments are listed below this letter.

We hope that our revised manuscript has been substantially improved and is now acceptable for publication in your journal and look forward to hearing from you soon.

With best wishes,

Yours sincerely,

Yuan Shi

Responses to Reviewer’ s Comments

  1. The authors repeat in the introduction all the wrong predictions of the experts from the early phase of the pandemic. In the introduction, the authors have to declare that many of our arguments to vaccinate children against COVID-19 have not come true from today’s perspective.

Response: We thank the reviewer’s careful reminders, and we really apologize for our careless mistake. We have deleted these erroneous views and mentioned in the Introduction that these erroneous views have not materialized today in our revised manuscript (L62-65).

  1. Nobody is interested in local reactions, fever, or headaches for a short time. By recognizing this, long-term, disabling side effects like Glomerulonephritis, Myocarditis, or chronic fatigue syndrome are of high importance. The question is why these important side effects are not recognized in so many expensive studies.

Response: Thank you for your thoughtful advice. What the reviewers said is very accurate, local reactions are not important, and long-term adverse reactions deserve more attention. However, we are sorry that there are almost no reports of glomerulonephritis, myocarditis, and chronic fatigue syndrome caused by COVID-19 vaccines. We have pointed out this shortcoming in the Discussion section and expect more researchers to focus on long-term adverse reactions after vaccination (L448-450).

Reviewer 3 Report

Thank you for inviting me to review this manuscript.
In this review the authors aim to synthesize the evidence for the safety, efficacy, and immunogenicity of varying types of COVID-19 vaccines in children younger than 18 years as an update to these previously performed systematic reviews.
I have enjoyed reading this manuscript and I found it very interesting and, even if I’m not an expert on the technical issues related to literature reviews, to me, the work seems well performed and sufficiently detailed.
I think that this review can be published in its current form.

Author Response

Dear Reviewer,

Thank you very much for your constructive comments and suggestions on our manuscript (Manuscript ID: vaccines-2100003). We have changed the name of the manuscript to “Safety, Efficacy, and Immunogenicity of Varying Types of COVID-19 Vaccines in Children Younger Than 18 Years: An Update of Systematic Review and Meta-Analysis”. In addition, our point-by-point responses to the comments are listed below this letter.

We hope that our revised manuscript has been substantially improved and is now acceptable for publication in your journal and look forward to hearing from you soon.

With best wishes,

Yours sincerely,

Yuan Shi

Responses to Reviewer’ s Comments

  1. The work seems well performed and sufficiently detailed. I think that this review can be published in its current form.

Response: We thank the reviewer for the encouraging comment!

Reviewer 4 Report

The manuscript “Evaluation of Covid-19 Vaccine in Children Younger Than 18 Years: An Update of Systematic Review and Meta-Analysis” is a review of different COVID-19 vaccines e for children younger than 18 years. A total of 17731 participants that received the COVID-19 vaccine and 7444 participants who received a placebo were included in this study ranging in age from 6 months to 17 years old. The topic of this work is appropriate for Vaccines, however, I listed a number of comments which I feel should be addressed.

Major comments:

-the title of the manuscript is generic but the aim is a more specific focus on the evaluation of “safety, efficacy, and immunogenicity of varying types of COVID-19 vaccines in children younger than 18 years”. In my point of view, the aim is much more adequate, and a specific title that covers the aim could improve the manuscript.

-Considering COVID-19 vaccines as mRNA, subunit, inactivated, and adenoviral vector vaccines, did you consider analyse safety, immunogenicity, and the efficacy COVID-19 vaccines comparing with placebo by type and with all types of vaccine included?

-Is the number of doses included in the analysis?

-The risk of bias appears much more positive. The presence of a different type of vaccines, different variants, and different analyses of safety, efficacy, and immunogenicity could be inserted in other bias? The adequate insertion of bias is imperative to meta-analysis.

-Could you indicate if is appropriate insert analysis on adolescents with 18 years old and 5 months-old? As occurred in Table 1. If you remove the study [32] the age range is between 3 to 18 years. In my point of view, it is more adequate because of the inclusion of babies and Children with ~18 Years is very different.

-It is critical to include in the discussion the most common adverse reactions reinforcing that severe adverse effects were not detected after Covid-19 vaccine application.

-There are inconsistencies in table 3, column 4 (age). If the study was performed in Children Younger Than 18 Years, some data as 18-25 (and others) appears wrong. If just the study of Anderson et al. includes children with less than 3 years old, it could be removed. In column 3, is the number 57 correct? Why much studies select 57 days?

-In my point of view, it is critical the inclusion of a sentence in the abstract focused on efficacy of the COVID-19 Vaccines related to table 10.

-Conclusion of abstract and final conclusion (topic 5). The sentence “The mRNA and subunit vaccines elicited robust binding antibody responses to the prototype SARS-CoV-2 and recent variants, including omicron variants, and provided satisfactory protection” should be improved to respond to the raised question of the aim. The sentence L-429-432 should address the question raised in the aims.

Author Response

Dear Reviewer,

Thank you very much for your constructive comments and suggestions on our manuscript (Manuscript ID: vaccines-2100003). We have changed the name of the manuscript to “Safety, Efficacy, and Immunogenicity of Varying Types of COVID-19 Vaccines in Children Younger Than 18 Years: An Update of Systematic Review and Meta-Analysis”. We have revised the manuscript accordingly, and all amendments are indicated by the red font in the revised manuscript (Please see the attachment). In addition, our point-by-point responses to the comments are listed below this letter.

We hope that our revised manuscript has been substantially improved and is now acceptable for publication in your journal and look forward to hearing from you soon.

With best wishes,

Yours sincerely,

Yuan Shi

Responses to Reviewer’ s Comments

  1. The title of the manuscript is generic, but the aim is a more specific focus on the evaluation of “safety, efficacy, and immunogenicity of varying types of COVID-19 vaccines in children younger than 18 years”. In my point of view, the aim is much more adequate, and a specific title that covers the aim could improve the manuscript.

Response: Thank you for pointing out this problem in our manuscript. We have changed the manuscript title which is indicated by the red font in our revised manuscript (L1-4).

  1. Considering COVID-19 vaccines as mRNA, subunit, inactivated, and adenoviral vector vaccines, did you consider analyze safety, immunogenicity, and the efficacy COVID-19 vaccines comparing with placebo by type and with all types of vaccine included?

Response: We thank the reviewer for pointing out this issue. What the reviewers said is very accurate, we have made the changes as suggested by the reviewer in our revised manuscript.

  1. Is the number of doses included in the analysis?

Response: Thank you for pointing out this issue in our manuscript, and we quite agree with your suggestion. We considered analyzing the different number of doses of COVID-19 vaccines, but unfortunately, we found that each age group received different doses of the vaccine in different studies in the mRNA (Ali 2021 et al., Anderson 2022 et al., Buddy Creech et al., Frenck et al., Walter et al.) and the inactivated (Han 2021 et al., Xia 2021 et al., Xia 2022 et al.) vaccine groups. In the subunit vaccine group, three RCTs (Anez et al., Liu et al., Thuluva et al.) provided information on recipients aged 3–17 years after receiving the same doses (0.5 mL) of subunit vaccine; and only one RCT (Zhu et al.) is about the adenovirus vaccine, and all recipients receiving 0.3 mL dose of vaccine. Therefore, analyses failed to be performed on this basis.

  1. The risk of bias appears much more positive. The presence of a different type of vaccines, different variants, and different analyses of safety, efficacy, and immunogenicity could be inserted in other bias?

Response: Thank you for your wise advice. That’s exactly right. The factors mentioned by the reviewer will cause bias. Therefore, we have conducted separate meta-analyses based on different included COVID-19 vaccine types, this leads to a significant decrease in bias. However, most of the included studies did not mention the locally prevalent variants at the time of the experiment, we were unable to perform subgroup analysis based on different variants, this is a limitation in our previous manuscript and has been mentioned in our revised manuscript (L440-442).

  1. Could you indicate if is appropriate insert analysis on adolescents with 18 years old and 5 months-old? As occurred in Table 1. If you remove the study [32] the age range is between 3 to 18 years. In my point of view, it is more adequate because of the inclusion of babies and Children with ~18 Years is very different.

Response: We are very grateful to the reviewer for pointing out this issue, and we think it is absolutely right. However, given the number of young cases that have drawn urgent attention in China as the country’s defenses against COVID-19 have weakened, it could be a sign that more young children will be vaccinated in the future. In addition, due to the overall heterogeneity of this study, we conducted a separate analysis of it. We found that in the safety analysis, among children under three years of age, younger children were safer to receive the COVID-19 vaccines, which was consistent with the results in the participants over three years of age; in the effectiveness section, we found that the COVID-19 vaccine did not protect children under three years of age as strongly as children over three years of age, which may be related to age and the fact that the study period was during the omicron epidemic. At the same time, the deletion of this literature will result in a huge workload. In the future, we will make a new Meta according to the newly published research and the reviewer's instructions.

  1. It is critical to include in the discussion the most common adverse reactions reinforcing that severe adverse effects were not detected after Covid-19 vaccine application.

Response: We thank the reviewer for pointing out these issues. We have corrected all the above problems in our revised manuscript (L375-377).

  1. There are inconsistencies in table 3, column 4 (age). If the study was performed in Children Younger Than 18 Years, some data as 18-25 (and others) appears wrong. In column 3, is the number 57 correct? Why many studies select 57 days?

Response: We thank the reviewer for pointing out this issue. Table 8 is the immunogenicity of the included studies. The immunogenicity was expressed through the noninferiority of the immune response. The noninferiority was reflected by comparing the geometric mean ratio (GMT) and the serologic responses in children and young adults (such as 18-25 and other ages). Notably, the data (as 18-25 and others) of immunogenicity in young adult participants were provided from previous experiments aimed at adults. The day 57 on behalf of 1 month after dose 2 of COVID-19 vaccines when the body’s antibody response is higher and more stable.

  1. It is critical the inclusion of a sentence in the abstract focused on the efficacy of the COVID-19 Vaccines related to table 10.

Response: We thank the reviewer for pointing out this issue in our manuscript. we have added a sentence describing the effectiveness of COVID-19 vaccines in the Abstract section (L26-27).

  1. Conclusion of abstract and final conclusion (topic 5). The sentence “The mRNA and subunit vaccines elicited robust binding antibody responses to the prototype SARS-CoV-2 and recent variants, including omicron variants, and provided satisfactory protection” should be improved to respond to the raised question of the aim. The sentence L-429-432 should address the question raised in the aims.

Response: Thank you for your thoughtful advice. we have made the changes as suggested by the reviewer in our revised manuscript (L453-457).

Round 2

Reviewer 1 Report

Thank you for resubmission.

In the re-submission, we believe that the answers to reviewer's comments  have been appropriately added and corrected to make them easier to understand.

Although data on adverse effects of vaccines on children and their effects need to be carefully accumulated in the future, I believe that this is a coherent discussion of the vaccines used at this time and provides important insights.

Reviewer 2 Report

Thanks a lot for this revision that sounds now more scientific and less political.